# Peptides That Block RAS-p21 Protein-Induced Cell Transformation

**DOI:** 10.3390/biomedicines11020471

**Published:** 2023-02-06

**Authors:** Matthew R. Pincus, Bo Lin, Purvi Patel, Elmer Gabutan, Nitzan Zohar, Wilbur B. Bowne

**Affiliations:** 1Department of Pathology, SUNY Downstate Medical Center, 450 Clarkson Avenue, Brooklyn, NY 11203, USA; 2AdventHealth, Department of Pathology, 301 Memorial Medical Pkwy, Daytona Beach, FL 32117, USA; 3Jefferson Pancreas, Biliary and Related Cancer Center, Department of Surgery, Thomas Jefferson University, 1015 Walnut Street, Curtis Building, Suite 618, Philadelphia, PA 19107, USA

**Keywords:** RAS-p21 protein, oncogenic forms, peptides, small molecules, amino acid substitutions, mutant protein, cell transformation, blockade of oncogenic protein

## Abstract

This is a review of approaches to the design of peptides and small molecules that selectively block the oncogenic RAS-p21 protein in ras-induced cancers. Single amino acid substitutions in this protein, at critical positions such as at Gly 12 and Gln 61, cause the protein to become oncogenic. These mutant proteins cause over 90 percent of pancreatic cancers, 40–50 percent of colon cancers and about one third of non-small cell cancers of the lung (NSCCL). RAS-p21 is a G-protein that becomes activated when it exchanges GDP for GTP. Several promising approaches have been developed that target mutant (oncogenic) RAS-p21 proteins in these different cancers. These approaches comprise: molecular simulations of mutant and wild-type proteins to identify effector domains, for which peptides can be made that selectively inhibit the oncogenic protein that include PNC-1 (ras residues 115–126), PNC-2 (ras residues 96–110) and PNC7 (ras residues 35–47); the use of contiguous RAS-p21 peptide sequences that can block ras signaling; cyclic peptides from large peptide libraries and small molecule libraries that can be identified in high throughput assays that can selectively stabilize inactive forms of RAS-p21; informatic approaches to discover peptides and small molecules that dock to specific domains of RAS-p21 that can block mitogenic signal transduction by oncogenic RAS-p21; and the use of cell-penetrating peptides (CPPs) that are attached to the variable domains of the anti-RAS-p21 inactivating monoclonal antibody, Y13 259, that selectively enters oncogenic RAS-p21-containing cancer cells, causing these cells to undergo apoptosis. Several new anti-oncogenic RAS-p21 agents, i.e., Amgen’s AMG510 and Mirati Therapeutics’ MRTX849, polycyclic aromatic compounds, have recently been FDA-approved and are already being used clinically to treat RAS-p21-induced NSCCL and colorectal carcinomas. These new drugs target the inactive form of RAS-p21 bound to GDP with G12C substitution at the critical Gly 12 residue by binding to a groove bordered by specific domains in this mutant protein into which these compounds insert, resulting in the stabilization of the inactive GDP-bound form of RAS-p21. Other peptides and small molecules have been discovered that block the G12D-RAS-p21 oncogenic protein. These agents can treat specific mutant protein-induced cancers and are excellent examples of personalized medicine. However, many oncogenic RAS-p21-induced tumors are caused by other mutations at positions 12, 13 and 61, requiring other, more general anti-oncogenic agents that are being provided using alternate methods.

## 1. Background

Ras oncogenes that encode the RAS-p21 protein (Mr 21kDa, 189 amino acid residues) are known to be important causative factors in a large number of human cancers. Over 90 percent of pancreatic cancers, 40–50 percent of colon cancers, and about one-third of non-small-cell carcinomas of the lung express the ras oncogene [1]. This oncogene also occurs in a significant number of non-solid tissue tumors, such as acute myelogenous leukemia. These oncogenes are mutant forms of the normally occurring ras gene that encodes the RAS-p21 protein, a G protein that is a major promoter of mitosis via mitogenic signal transduction pathways, among other important cell functions, such as differentiation, migration or apoptosis.

### 1.1. The Ras-Gene-Encoded p21 Protein

Most of the mutations involve single base changes in the first exon of the ras gene most commonly at position 12 resulting in a single amino acid substitution for the normally occurring glycine at this position [2]. Otherwise, the normal ras gene and the ras oncogene are the same. Transfection of the ras oncogene into untransformed fibroblasts, NIH 3T3 cells, results in malignant cell transformation while transfection of the normal gene into these cells has no effect on these cells [2]. Importantly, microinjection of the cloned purified RAS-p21 protein with a Val for Gly 12(G12V) substitution into NIH 3T3 cells induces cell transformation, while injection of the wild-type (Gly 12-containing) protein has no effect on these cells [3]. Thus, the ras oncogene-encoded p21 protein is the cause of cell transformation, and a single amino acid substitution of Val for Gly 12 causes the RAS-p21 protein to become oncogenic. Further studies revealed that any RAS-p21 protein that contains the substitution of any non-cyclic L-amino acid for Gly 12 is oncogenic; only the substitution of Pro for Gly 12 results in a non-transforming protein [2]. Similar studies [2] further demonstrate that, with the exception of Glu and Pro, arbitrary amino substitutions for Gln 61 (Q61) result in an oncogenic protein. Other more limited numbers of substitutions such as at Gly 13 and several other residues in RAS-p21 also result in oncogenic proteins.

Importantly, it was shown that the microinjection of oncogenic (G12V)-RAS-p21 into frog (*Xenopus laevis*) oocytes, that are metaphase-arrested cells in the second meiotic division, induces completion of the second division, resulting in oocyte maturation [4]. An identical effect was demonstrated by incubating these oocytes with insulin [4]. It was shown that the microinjection of the neutralizing high affinity anti-RAS-p21 antibody, Y13 259, into oocytes that were then incubated with insulin, blocked maturation, thus demonstrating that insulin induced oocyte maturation by the activation of wild-type RAS-p21 [5]. As a result, oocytes proved to be an excellent screening system for agents that block oncogenic RAS-p21 but not its activated wild-type counterpart protein.

### 1.2. Mitogenic Signal Transduction Pathways Induced by RAS-p21

There are three major forms of RAS-p21: H (Harvey)-ras-, K (Kirsten)-ras- and N (neuroblastoma)-RAS-p21. H-RAS-p21 contains 189 amino acid residues (the others have a few more or less residues) and differ in sequence from one another, mostly at the carboxyl terminal end of the protein. All forms are G-proteins, i.e., they bind to GDP and become activated when they exchange GDP for GTP. This event must occur in the cell membrane where the RAS-p21 protein can interact with proteins described below that link RAS-p21 to growth factor receptors that span the cell membrane [1]. To induce the membrane localization of RAS-p21, the protein is farnesylated at Cys 186 wherein the -SH group of Cys 186 displaces the PPi moiety of farnesyl pyrophosphate catalyzed by the enzyme farnesyl transferase, a target for potential therapeutics. Cys 186 occurs on the carboxyl terminal end of RAS-p21 in the membrane localization “CAAX box” sequence of the form CAAX where C is Cys, A is a hydrophobic amino acid and X is any amino acid. Often, a Lys residue precedes the CAAX box. The activation of membrane-bound RAS-p21 involves several steps, as shown in Figure 1. In the first step, a growth factor, such as epidermal growth factor (EGF) or insulin, binds to its growth factor receptor, which contains three domains: the extracellular growth factor-binding domain, the transmembrane domain and the intracytosolic domain. Upon binding to its growth factor, the growth factor receptor dimerizes, leading to the activation of a tyrosine kinase domain on the cytosolic domain of the receptor. Upon activation, the tyrosine kinase domain phosphorylates tyrosine residues in the cytosolic domain of the receptor, allowing the receptor to bind to SH2 domains of adaptor proteins such as GRB2 [6], leading to the membrane localization of guanine exchange protein (GEF) or guanine nucleotide exchange proteins (GNEF). The latter two proteins bind to RAS-p21 and cause its activation by inducing the exchange of GDP for GTP. An important GNEF is called SOS (son of sevenless, a similar protein identified in Drosophila) [7]. Opposing these events, GTPase-Activating Proteins (GAP), a family of proteins, bind to the RAS-p21-GTP complex and induce inherent GTPase activity, resulting in the hydrolysis of GTP to GDP, presumably causing the complex to adopt an inactive form [8].

Once activated, RAS-p21 initiates a series of signal transduction pathways from the cell membrane to the nucleus, resulting in mitosis. In a critical signal transduction pathway [1], RAS-p21 binds to the critical serine/threonine kinase protein, RAF, which, upon binding to RAS-p21, activates a second critical serine/threonine kinase called MEK that, in turn, activates the all-important kinase, ERK (1 and 2), also called MAPK (extracellular mitogenic-activating protein kinase or microtubular-activating protein kinase), which upon activation, crosses the nuclear membrane and activates the vital pro-mitogenic transcription factor, fos. Activated fos forms a heterodimeric complex with the transcription factor, jun, which is activated by another kinase called jun kinase (JNK) that can be directly activated by oncogenic RAS-p21 (see below). As discussed further, the RAS-p21–RAF interaction is a site that can be targeted by a number of peptides that block this interaction. The fos–jun complex activates the transcription of pro-mitotic proteins including the cyclins and nuclear matrix proteins (NMPs in Figure 1) required for nuclear skeleton maintenance during mitosis.

As also shown in Figure 1, there are a number of other pathways activated by RAS-p21. Activated RAS-p21 binds to and activates phosphoinositol-3-hydroxykinase (PI3K) [1], causing a direct increase in intracellular phosphatidylinositol triphosphate (PIP3) that is involved in the activation of protein kinase B (Akt), of which promotes mitogenesis (not shown in the figure). PI3K also activates the mTOR protein that is involved in multiple cell functions, including cell growth and proliferation (not shown in Figure 1).

RAS-p21 also activates phospholipase C (PLC) [9], which results in the synthesis of diacylglycerol (DAG) that activates protein kinase C (PKC); this was found to be essential for mitogenic signal transduction by oncogenic RAS-p21 [10] and inositol triphosphate, of which induces the mobilization of calcium ions that are critical in the kinase activation cascade.

Importantly, it was found that both activated wild-type RAS-p21 and Val 12-p21 interact with the protein jun kinase (JNK) and the amino terminal regulatory domain (residues 5–89) of its target, jun, in in vitro binding experiments. These interactions result in the phosphorylation and activation of JNK [11]. The binding and phosphorylation of JNK were enhanced by the addition of GTP but not GDP to the assay systems [11]. Val 12-RAS-p21 was found to show enhanced binding to JNK and jun and enhanced phosphorylation of JNK that was five- to ten-fold higher for oncogenic RAS-p21 than for the wild-type protein [11]. As discussed below, JNK activation appears to be critical for the oncogenic RAS-p21 pathway. This is indicated on the right side of Figure 1.

### 1.3. Structure of RAS-p21

The X-ray crystal structures of wild-type and oncogenic forms of RAS-p21 have been determined (12,13). In the crystal structure of RAS-p21 bound to the activating non-hydrolyzable GTP analogue, the ϒ-phosphoroamidate form of GTP (GppNHp), there are distinct domains that interact with this activating ligand [12]. These include the P loop domain, containing residues 4–20 that contain the critical mutational sites including Gly 12 and Gly 13; the switch 1 domain, containing residues 32–47; an effector domain, of which has been shown to be involved in the interaction of RAS-p21 with critical downstream targets such as RAF discussed in the next section below; and the switch 2 domain, consisting of residues 55–76 and containing other critical mutational sites such as Gln 61 and Gly 60. This domain contains a short α-helical segment containing residues 67–73, called the α2 helix, which can interact with another α-helical domain, the α3 helix, involving residues 89–103. The disposition of the two switch domains and the α3 helix can give rise to drug-accessible pockets, enabling specific agents, including peptides, to lock oncogenic RAS-p21 in inactive states, mainly affecting the conformation of the switch 1 domain, resulting in the blockade of uncontrolled cell division, as discussed in ensuing sections.

In the structure of the activated wild-type RAS-p21 bound to GppNHp, guanine ring and pendant atoms form hydrogen bonds with Asp 119, Ser 145 and Ala 146; the ribofuranose ring forms hydrogen bonds with Val 29 and Asp 30 and Lys 117. Importantly, all three phosphate residues form a network of hydrogen bonds with P-loop residues, i.e., in the domain involving residues 4–20: the α-phosphate oxygens form multiple hydrogen bonds with the main chain and -OH side chain of Ser 17 and main chain of Ala 18, while the connecting α-β oxygen in this phosphate forms a hydrogen bond with the backbone of Gly15. Val 14, Gly 15, Ser 17 and Lys 16 all form hydrogen bonds to the β-phosphate atoms. Gly 13 forms a hydrogen bond to the β-ϒ nitrogen of the ϒ-phosphoroamidate nitrogen. Interestingly, residues from the P-loop, switch 1 and switch 2 domains (the latter containing residues 59–64 called the L4 loop) form extensive hydrogen bonding interactions with the ϒ-phosphate including Lys 16, Tyr 32, Thr 35 and Gly 60. All three of these domains are involved in the critical conformational changes that occur in the activation of RAS-p21.

### 1.4. Effects of GTP Binding and Oncogenic Amino Acid Substitutions on the Structure and Function of RAS-p21

As noted in Section 1.1 above, arbitrary non-cyclic-L-amino acid substitutions for Gly 12 cause the RAS-p21 protein to become oncogenic. A comparison of the X-ray crystal structures of wild-type RAS-p21 with oncogenic forms of RAS-p21 including G12R, G12V, Q61His, Q61L between 2 and 2.6 Å resolution indicated no major structural changes between the wild-type and mutant forms [13]. Rather, each amino acid substitution at both the 12 and 61 positions could likely cause the loss of GTPase activity due to a disruption of the positioning of functional groups (like the carboxamido group of Gln 61) that would be involved in GTP hydrolysis, resulting in the prolonged activation of the mutant protein. For example, hydrogen bonding of the backbone NH of Gly 12 to a ϒ-phosphate oxygen might be disrupted by a side chain of a substituted amino acid while the loss of the carboxamido group of Gln 61 would prevent a similar type of interaction with this phosphate moiety in the RAS-p21 active site. In addition, these substitutions were considered to lower GAP activity in promoting β-ϒ-phosphate hydrolysis [13].

On the other hand, Pro 12, which has no backbone NH, is non-oncogenic. Additionally, the ring atoms of Pro would appear to be at least as bulky as the methyl side chain of Ala at this position, which is an oncogenic amino acid substitution. In addition, the substitution of Gly for Gln 61 is oncogenic but has high GTPase activity. Other mutated RAS-p21 proteins such as D38E RAS-p21 bind with high affinity to GAP and to GTP but are enzymatically inactive, resulting in a low GTP hydrolysis rate; however, they are non-transforming. The mutant A59T RAS-p21 protein has at least five times the GTPase activity of wild-type RAS-p21 but transforms NIH3T3 cells with as high a rate as GTPase-defective oncogenic RAS-p21 proteins such as G12R- and G12K RAS-p21. Even more significantly, two triply substituted RAS-p21 mutant proteins have been identified that do not bind to GDP or GTP at all: The first is [G10V,G12R,A59T]RAS-p21, which transforms cells with a high efficiency, and the other is [G12R,G15V,A59T]RAS-p21 which is non-transforming [14]. Since neither protein binds to a nucleotide, the difference in activity between these two proteins cannot be due to differences in GTPase activity, but must be due to structural differences.

In line with this conclusion, conformational studies [15] have indicated that critical amino acid substitutions at positions 12 and 61 can cause significant local changes in domain conformation, such as in switch 1 and switch 2 segments, and have been used to design specific peptides from the RAS-p21 protein itself that inhibit the oncogenic form of RAS-p21 selectively, which therefore may be of use in the treatment of ras-induced tumors.

### 1.5. Peptides and Small Molecules Are Promising Agents That Can Selectively Destroy Oncogenic RAS-p21-Induced Cancers

In the next three sections, we present four ways in which specific peptides and small molecules block oncogenic RAS-p21, making them highly useful in treating RAS-p21-induced tumors. These include: 1. the identification of peptides corresponding to domains of RAS-p21 that selectively inhibit oncogenic RAS-p21 using molecular simulations and peptide library approaches in which overlapping peptide sequences from RAS-p21 are assayed for their abilities to block oncogenic RAS-p21 selectively; 2. a cyclic peptide library approach that generates peptides based on their abilities to bind to grooves in G12D-RAS-p21 that stabilize an inactive form of this protein bound to GTP, and a small molecule library screening approach for poly-aromatic compounds that can insert into a groove present specifically in the GDP-bound form of G12C-RAS-p21 that permanently locks this oncogenic form of RAS-p21 in its inactive GDP-bound state; 3. an informatic approach in which candidate peptides are computed for their abilities to dock to specific domains of wild-type and oncogenic forms of RAS-p21 and the concurrent use of a small molecule library to generate candidate analogues of GTP; and 4. a novel immunologic technique in which a peptide that targets cancer cell membranes is fused to the anti-RAS-p21 antibody Y13 259 that inactivates oncogenic RAS-p21.

## 2. Anti-Oncogenic RAS-p21 Peptides from RAS-p21 Itself

To determine the possible structural effects of oncogenic amino acid substitutions in RAS-p21 on its three-dimensional structure, conformational energy calculations, using two types of sampling methods, i.e., molecular dynamics (using AMBER [16] and DISCOVER [17] potential functions) and the electrostatically driven-Monte Carlo (EDMC) [18] method (using ECEPP potential functions [19]), were performed on the structures of wild-type-RAS-p21 and G12V- and Q61L-RAS-p21 (oncogenic forms) bound to GDP and GTP. These procedures were likewise applied to the two triple mutant proteins [G10V,G12R,A59T]RAS-p21 and [G12R,G15V,A59T]RAS-p21 that do not bind to either GDP or GTP [14]. Both methods determined the low energy structures that lie in the same potential energy well as the energy-minimized X-ray starting structures which have the same chain fold but may differ locally in conformation from this starting structure. These low energy Boltzman-weighted structures, including that for the energy-minimized X-ray starting structure, were then used to compute the average structures for the wild-type and oncogenic proteins which were then superimposed on one another. Local domain changes in conformation could then be readily determined from these superpositions. Under the hypothesis that these domains were potential effector peptide segments activated by the oncogenic amino acid substitutions, these peptides were then synthesized and tested in cells to determine if they could act as decoy peptides, i.e., as inhibitors of oncogenic RAS-p21 [15].

Both sets of approaches identified five highly flexible domains that underwent local structural changes in oncogenic RAS-p21 forms when their average structures were compared with that of the wild-type protein bound to GDP: the 10–16 sequence of the P loop domain containing residue 12, the switch 1 region; residues 35–47, a segment that interacts directly with the ras-binding domain (RBD) of RAF, and with GAP, SOS and PI3K proteins; the switch 2 region involving residues 55–71; and residues 96–110 and 115–126, both found to be involved the binding of RAS-p21 to SOS [15]. The 96–110 segment has also been implicated in the interaction of Val 12-RAS-p21 with JNK [15]. The C^α^ traces of each of these segments are shown in Figure 2 for the average structures of wild-type-RAS-p21-GDP (purple), wild-type RAS-p21-GTP (yellow), G12V-RAS-p21-GTP (blue) and Q61L-RAS-p21-GTP (green). Further correlation analysis on the structural changes in the different domains was performed that indicated that changes in conformation in the 10–16 domain around residue 12 are associated with structural changes in the switch 1 region which, in turn, are associated with changes in the switch 2 domain. Structural changes in the switch 2 domain were further found to correlate with changes in the 96–100 and 115–126 domains [20].

### 2.1. Three RAS-p21 Peptides Selectively Block Oncogenic RAS-p21

Based on these results, peptides corresponding to these domains were synthesized and tested for their abilities to inhibit oncogenic RAS-p21 in *Xenopus laevis* oocytes. Three peptides were found to block microinjected G12V-RAS-p21 but were found to cause low or absent inhibition of insulin-induced oocyte maturation, i.e., 35–47 (PNC-7, TIEDSYRKQVVID), 96–110 (PNC-2, YREQIKRVKDSDDVP) and 115–126 (PNC-1, GNKCDLAARTVE). All three of these peptides were further tested against progesterone-induced oocyte maturation which, in contrast to insulin, is known to occur by ras-independent pathways [15], and none of these peptides had any effect on progesterone-induced maturation, supporting their specificities for ras-dependent pathways.

#### 2.1.1. Site of Action of PNC-7 (RAS-p21 Residues 35–47)

Since PNC-7 is from the switch 1 domain of RAS-p21 and known to interact with RAF as a downstream target, to localize its site of action, full length c-RAF was injected into oocytes and found to induce maturation that was blocked by PNC-7. In contrast, an oncogenic RAF form, RAF BXB, that has its amino terminal regulatory domain deleted, induced oocyte maturation that was not inhibited by PNC-7 [15]. These results suggest that the site of inhibition of PNC-7 was the amino terminal regulatory domain of RAF. In view of the finding that microinjection into oocytes of a construct expressing dominant negative RAF (dn-RAF) inhibited both oncogenic RAS-p21- and insulin-induced oocyte maturation, there are further suggestions that oncogenic and wild-type RAS-p21 interact with RAF differently and may activate oocyte maturation by different pathways [15].

Evidence has accumulated that supports this conclusion. In oocytes injected with oncogenic RAS-p21, there was a dramatic increase in the level of phosphorylated (activated) MAPK (ERK) and jun kinase (JNK) [15,21], presumably stimulated by activated RAF in contrast to levels expressed in oocytes induced to mature with insulin which were significantly lower than in the oncogenic RAS-p21-matured oocytes, despite the fact that the expression of both downstream proteins was the same in both sets of oocytes. These studies concluded that the binding of oncogenic RAS-p21 to RAF resulted in high levels of phosphorylated MAPK (MAPK-P), and that activated wild-type RAS-p21, while it utilized the RAF-MEK-MAPK pathway to some extent, did not require this pathway since there are other RAF targets that were activated by the wild-type RAS-p21–RAF complex [21].

##### Alternate Targets of RAF

Several of these putative targets of activated wild-type RAS-p21–RAF complexes not utilized by oncogenic RAS-p21 proteins were subsequently identified, using differential gene expression libraries for oocytes that were induced to mature either with oncogenic (Val 12)-ras p21 or insulin. In the wild-type RAS-p21-matured oocytes but not in the oocytes induced to mature with Val 12-RAS-p21, two dual specificity kinases, TOPK (T-Cell Origin Protein Kinase) and Dyrk1A (Dual Specificity Tyrosine Phosphorylation Regulated Kinase 1A), were expressed [22]. The former is in the MEK family and is known to interact with RAF. siRNA against the expression of each of these proteins were injected into oocytes that were induced to mature either with Val 12-p21 or with insulin and were found to block maturation only in the *insulin-treated* oocytes [15]. These findings suggested that both proteins participate on the signal transduction pathway of the activated wild-type protein but not the oncogenic protein.

Molecular dynamics calculations on the structures of wild-type- and G12V-RAS-p21 bound to the ras-binding domain (RBD) of RAF (residues 25–131) show that there are major shifts in domain positions in the RBD, especially in a surface loop involving residues 97–110 of the RBD [15], as shown in Figure 3. This peptide has been synthesized and found to block G12V-RAS-p21-induced oocyte maturation with no effect on insulin-activated wild-type RAS-p21-induced maturation [15], suggesting that the interactions between the two ras proteins and RAF differ.

#### 2.1.2. Site of Action of PNC-1 (GNKCDLAARTVE) and PNC-2 (YREQIKRVKDSDDVP)

Neither peptide was found to inhibit c-RAF-induced oocyte maturation, excluding RAF as a site of action for these peptides. In separate studies, oncogenic RAS-p21 was found to bind to jun-N-terminal kinase (JNK) that activates the transcription factor, jun (see Figure 1) with an affinity that was at least five times that for the wild-type protein [11,23]. The dose–response curve for PNC-2 for its inhibition of oncogenic RAS-p21-induced oocyte maturation was superimposable on the dose–response curve for its blockade of the binding of oncogenic RAS-p21 to JNK-jun, suggesting that the site of inhibition was in the formation of the oncogenic RAS-p21-JNK-jun complex [15]. Compatible with this conclusion was the finding that PNC-2 inhibits the phosphorylation of JNK in oocytes injected with oncogenic RAS-p21 [21].

### 2.2. Effects of PNC-2 and PNC-7 in Cancer and Normal Cells

Both PNC-2 and PNC-7 were synthesized and attached on their carboxyl termini to a leader transmembrane-penetrating sequence (KKWKMRRNQFVKVQRG) from *Drosophila* antennapedia protein that enables the peptides to be transported across cell membranes—referred to as cell-penetrating peptides or CPP [24]. As discussed in ref. [24] and references therein, CPPs are short peptides or, in some cases, whole proteins such as the drosophila *Antennapaedia* protein, that cause peptides or whole proteins or other types of molecules bound to them, usually covalently but sometimes non-covalently, to cross cell and nuclear membranes. After entry into the cell, the CPPs are often cleaved, allowing for their cargo to exert their effects intracellularly. Generally, CPPs contain a high percentage of positively charged amino acid residues, such as Lys, Arg and sometimes His. Other types of CPPs contain a predominance of non-polar, hydrophobic amino acids such as Leu, Ile,Val, Phe and Met or alternating segments of polar and non-polar residues. The mechanisms by which trans-membrane penetration occurs are largely unknown but are thought to involve endocytosis, using a variety of possible mechanisms such as pore formation from interactions of the positively charged amino acids with negatively charged membrane moieties such as phospholipids and heparan sulfates and formation of inverted micelles, among a number of other proposed mechanisms.

These two modified peptides were incubated with a normal pancreatic acinar cell line (BMRPA1) and its k-ras (G12V) transformed counterpart cell line TUC-3. The transplantation of TUC-3 but not BMRPA1 cells in nude mice resulted in metastatic pancreatic cancer with peritoneal carcinomatosis [25]. Incubation of these peptides with TUC-3 cells resulted in complete phenotypic reversion. The same results were obtained on human fibrosarcoma (HT1080) cells [25,26]. Neither peptide had any effect on the growth or viability of BMRPA1 normal acinar cells, indicating that the effects of these peptides were specific to cancer cells. Interestingly, both peptides induced the tumor cell necrosis of MIA-PaCa-2 cells, a human metastatic pancreatic cancer cell line, homozygous for the oncogenic G12C mutation (see below) [25].

These results were attributed to the zygosity of the different cell lines. Both TUC-3and HT1080 cells are heterozygous for oncogenic RAS-p21. It was postulated that, if a blockade of the oncogenic pathway occurs, a normal RAS-p21 protein can utilize alternate pathways, resulting in reversion to the untransformed phenotype. Additionally, since MIA-PaCa-2 cells are homozygous for oncogenic RAS-p21 and do not have normally functioning RAS-p21, a blockade of the oncogenic pathway blocks cell growth, resulting in cell death. In all cancer cell lines treated with either peptide, JNK-P levels were continuously reduced as a function of time, as found in the oocyte studies [15,25]. These results suggest that these peptides may be effective in treating RAS-p21-induced cancers with no off-target effects on normal cells.

### 2.3. Peptides from Proteins That Interact Directly with RAS-p21 or Selectively Block Targets on the Oncogenic RAS-p21 Pathway

A comparison of the average structures for G12V-RAS-p21-GppNH-p and wild-type RAS-p21-GppNHp bound to the ras-binding domain (RBD) of RAF revealed a major 90° flip in a surface loop involving in the RBD at its 97–110 sequence, as shown in Figure 3. The peptide corresponding to this domain (AVFRLLHEHKGKKA) was found to inhibit G12V-RAS-p21 selectively in the oocyte system. This result supports the concept that wild-type and oncogenic RAS-p21 proteins interact differently with RAF. Similarly, comparisons of the average structures from molecular dynamics applied to complexes of wild-type and G12V-RAS-p21 proteins bound to GAP and SOS [15] have revealed domains of these proteins that differ in structure. Peptides corresponding to these domains were synthesized and tested in the oocyte system [15]. From these studies, the peptide corresponding to SOS residues 994–1004 (LNPMGNSMEKE) selectively blocked G12V-RAS-p21-induced oocyte maturation.

In separate studies, the enzyme GST-π, which converts xenobiotic compounds into thio ether conjugates of glutathione, has been found to bind to JNK-jun complexes [27], and once bound, to block JNK-induced phosphorylation of jun [27]. Since JNK activation can occur directly by G12V-RAS-p21 (see above), GST-π was tested for its ability to block oncogenic RAS-p21 selectively. This protein was found to block oncogenic RAS-p21 protein but not insulin-induced oocyte maturation [15], further supporting JNK-jun as a unique target for oncogenic RAS-p21 [15]. Molecular dynamic studies on active GST-π and an inactive form enabled the identification of specific domains of this enzyme that might be involved in the binding of GST-π to JNK [15]. Six domains were identified, and peptides from these domains were synthesized and tested in the oocyte system [15]. Of these, two peptides, 34–50 (TIDTWMQGLLKPTCLYG) and 169–182 (CLDNFPLLSAYVAR), were found to block G12V-RAS-p21-induced oocyte maturation exclusively [15], suggesting that they may be effective in inhibiting oncogenic RAS-p21 selectively.

### 2.4. Use of Overlapping Peptides from RAS-p21 to Target Ras-RAF Interactions

Since a major signal transduction pathway activated by RAS-p21 involves the binding of RAS-p21 to RAF at the cell membrane, resulting in the downstream activation of MEK and ERK, agents have been devised to block this interaction. As an alternative to using conformational analysis to design ras peptides that could block this interaction as described above, the use of overlapping 12-mer (dodecapeptides) from the RAS-p21 protein was introduced. These peptides were then assayed for their abilities to serve as decoys to RAF, causing the displacement of oncogenic RAS-p21 from binding to RAF [28]. These peptides overlapped from the linear sequence of K-ras by two residues. For example, the first two peptides were residues 1–12 and 11–22, etc. This resulted presumably in the synthesis of eighteen dodecamers that were then bound to a membrane with which activated RAF was incubated, and the binding of RAF to these peptides was assayed via chemiluminescence assay. A contiguous string of peptides from the carboxyl terminus of K-RAS-p21 was found involving residues 169–188, including the CAAX box sequence at the carboxyl terminal end. Interestingly, this domain, called the hypervariable domain of RAS-p21, is the only one for which there are significant differences in the amino acid sequences among the various forms of RAS-p21. This domain is apparently highly flexible since its structure cannot be determined by X-ray crystallography [15]. Since it does contain the CAAX sequence that is farnesylated to enable the membrane attachment of RAS-p21, this peptide may block this attachment.

The RAS-p21 169–188 peptide was synthesized with a cell-penetrating peptide or CPP (leader sequence) on its amino terminus, termed Mut3DPTSh1, the whole sequence being: *VKKKIKAEIKI***KMSKDGKKKKKKSRTRCTVM**, called Mut3DPTSh1-Ras, where the italicized sequence is the CPP and the bold sequence is K-ras residues 169–188. This peptide was shown to compete for the binding of ras to RAF in MDA-MB-231 breast cancer cells [28]. Unfortunately, other peptide “hits” were found but were not assayed [28]. This consideration would include peptides such RAS-p21 residues 31–42 and 41–52 corresponding to peptide segments from the switch 1 domain that is known to interact with RAF [15]. Results for negative control peptides were not reported.

Mut3DPTSh1-Ras was then assayed for its ability to induce cancer cell death in a number of different cancer cell lines and was found to induce significant apoptosis in MDA-MB-231 breast cancer, SW626 ovarian cancer, SW480 colon cancer and HBCx17 breast cancer cell lines, but only minimally affected the cell viability of HCT-116 and HT-29 colon cancer cell lines, even though they carry K-ras mutations [28]. Peculiarly, though this peptide was shown as causing significant apoptosis of SW480 cells, it was found not to affect viability of these cells in the MTT cell viability assay [28]. This peptide also was found to induce a significant reduction in cell viability of H1650, HBEC wt and HBEC Ras V12, the former two lines being non-small cell lung cancer cell lines that have wild-type ras [28]. These findings suggest that this peptide does not specifically affect mutant ras and may even function in a ras-independent fashion.

Mut3DPTSh1-Ras was also evaluated on human chronic lymphocytic leukemia (CLL) cells from CLL patients as well as peripheral blood mononuclear cells from healthy donors [28]. There was marginal, if any, effect, specifically on CLL cells comparing to normal cells from healthy donors. In addition, there was a baseline 40% of apoptosis observed in normal cells with the treatment of this peptide, suggesting it may be toxic to normal cells [28].

This peptide was further tested in nude mice against hematopoietic tumor cell lines of B-cell origin, such as Burkitt’s lymphoma (Daudi cell line) and hairy cell leukemia (Jok 1 and 5.3 cell lines), the former of which the study claims is chronic lymphocytic leukemia-like [28]. None of these cell lines are known to express ras gene mutations, although the claim is made that Daudi cells express mutant ras. The Mut3DPTSh1-Ras peptide was found to prolong survival with high significance in mice bearing Jok.1 cells but not in Jok 5.3 cells, even though these cells do not express mutant RAS-p21. It was also found to inhibit tumor growth in nude mice of Daudi cells, but this effect was diminished in mice in which the tumors were permitted to grow prior to treatment.

The investigators in this study attempted to contrast their Mut3DPTSh1-Ras peptide results with those concerning the peptides employed to block oncogenic RAS-p21 selectively based on the results of molecular modeling as discussed in Section 2.1 above, stating that the results with such peptides as PNC-2 and PNC-7 (mislabeled as “PCN”) are “speculative” [28]. As noted in Section 2.1.2 above, both peptides strongly and selectively inhibited oncogenic (Val 12-) RAS-p21-induced oocyte maturation, and direct evidence indicates that PNC-7 blocks RAS-p21–RAF interactions by binding to the N-terminal regulatory domain of RAF, that wild-type and G12V-RAS-p21 interact differently with RAF while PNC-2 (and PNC-1) blocks the interaction of oncogenic RAS-p21 with JNK. Direct evidence that these latter two peptides block the binding of oncogenic RAS-p21 to JNK was provided by the finding that the binding of Val 12-RAS-p21 to bead-bound p-Gex-JNK and jun is strongly inhibited by the PNC-1 and PNC-2 peptides, such that the dose–response curve for the inhibition of binding superimposes on that for the selective inactivation of G12V-RAS-p21. All three peptides induced either phenotypic reversion in cancer cells heterozygous for mutant RAS-p21 or tumor cell necrosis of cancer cells that were homozygous for ras mutations. None of these peptides affected normal cells, suggesting that they are good candidates for treating cancers in vivo.

## 3. Anti-RAS-p21 Peptides from Expression Libraries

As noted in the Introduction section, X-ray crystallographic structures of wild-type and oncogenic RAS-p21 bound to GDP and GTP suggested that there is an absence of surface-exposed clefts to which drug molecules can bind, suggesting that RAS-p21 may be “undruggable [12,13]”. This impression has proved false by a series of studies that have resulted in a number of cyclic peptides [29,30] and small molecules that bind to mutant RAS-p21 proteins in their switch 2 domains in a binding “pocket” involving the switch 2 loop (residues 55–67) and the α-2 helix (residues 67–73) [31]. This binding is very much dependent on the specific amino acid substitutions that occur. When these agents bind, they strongly inhibit either GTP-bound RAS-p21 or GDP-RAS-p21. In the former case, GTP-bound RAS-p21 cannot bind to the RBD of ras, and in the latter case, GDP-bound RAS-p21 is “frozen” in an inactivated state such that it cannot exchange GDP for GTP and, since it cannot bind to GTP, it cannot adopt an activated conformation in which it presumably interacts with RAF.

### 3.1. Cyclic Peptides Freeze Oncogenic G12D-RAS-p21 in an Inactive GTP-Bound State

X-ray crystallographic studies on the G12D-RAS-p21 mutant protein bound to the non-hydrolyzable GTP analogue, GppNHp, indicated that a pocket (called P2) was present between the structure of residues 55–65 of the switch 2 domain and the α-2 helix involving residues 67–73 and was not present in this RAS-p21 protein bound to GDP [29]. This finding showed that at least this site was “druggable” and that possibly peptides or small molecules could be found that would bind to the pocket such that the GTP-bound state of G12D-RAS-p21 would be inactivated and would not interact with the downstream target, RAF.

Prior X-ray crystallographic and ^31^P-NMR studies on mutant T35S-RAS-p21 bound to GppNHp indicated that there were in fact two bound conformational forms called state 1 and state 2. In state 1, the switch 1 domain is not well-defined crystallographically and does not bind to the RBD of RAF (see above) [30]. In state 2, this domain adopts a conformation that results in the binding of this mutant ras protein to RAF [30]. Thus, the possibility existed that a drug could bind in the pocket area of G12D-RAS-p21-GppNHp such that it would freeze the complex in the state 1 inactive conformation, resulting in the selective inactivation of the G12D-RAS-p21 protein.

To this end, a high diversity cDNA expression library of over 10^12^ encoded compounds was employed [29], and a series of cyclic peptides was generated and tested for their abilities to block the binding of a double mutated G12D/T35S-RAS-p21 to the RAF RBD. Peptides that satisfied this condition were excluded if they also bound to the GDP-loaded double mutant protein. This process resulted in the selection of three cyclic dodecapeptides, each with an exocyclic Gly residue labeled as KD1, KD2 and KD3. Of these, KD2, shown in Figure 4, was found to have the highest relative affinity for binding to GppNHp-bound mutant RAS-p21. KD2 strongly inhibited the binding of G12D K-RAS-p21 to the RBD (IC_50_ = 12.5 uM) and blocked the SOS-mediated nucleotide exchange of the GDP-loaded protein. On the other hand, KD2 did not block wild-type RAS-p21 charged with GppNHp from binding to the RAF RBD, suggesting that this cyclic peptide will block only oncogenic G12D-RAS-p21 and would therefore not affect normal cells.

The X-ray crystal structure of KD2 bound to G12D-RAS-p21-GppNHp has been determined [29] and superimposed on the structures of G12D-RAS-p21-GppNHp and G12D-RAS-p21-GDP. The striking feature of these comparisons is that in the KD2-bound structure, there is a major movement of the α-2 helix by about 40° in the switch 2 domain and the peptide binds deeply in this groove between this α-2 helix and the switch 2 loop [29]. This groove is not seen in any of the X-ray structures of RAS-p21 bound to GppNHp. KD2 makes a number of favorable contacts in the groove with Gly 60 (switch 2), Asp 69 (switch 2 and α-2 helix) and Gln 99 (α-3 helix). In addition, the Thr 10 residue of KD2 interacts through a water molecule with Asp 12. Importantly, these changes do not occur in the structure of KD2 bound to G12D-RAS-p21-GDP. The segments involved in the unique groove and the structure of KD2 bound in the grove are shown in Figure 5. While the complete structure of the switch 1 domain involved in the binding of RAS-p21-GppNHp to the RAF RBD could not be determined, this domain was found to undergo a large movement (of up to 25 Å) in the KD2-bound protein compared with either the same domain of unliganded G12D-RAS-p21 or G12D-RAS-p21-GDP. Another significant movement was found also in the switch 2 domain (of up to 10 Å) [29].

Other cyclic peptides containing substitutions for Thr 11 in KD2 were found to have an enhanced effect on the binding of the peptides to G12D-RAS-p21-GppNHp. In particular, the substitution of an unnatural amino acid with a 4-methyl piperidine ring (called Aza X) for Thr 11 was found to have about a fifteen-fold decrease in its IC_50_ compared with that for KD2, although these cyclic peptides were found to block wild-type RAS-p21 at higher concentrations (100 Um range). In addition, bicyclic forms of the cyclic peptide, such as the disulfide bicycle in which two SH groups at positions 3 and 9 form a disulfide bridge, have been synthesized and found to block the binding of G12D-RAS-p21-GppNHp to the RAF RBD with a ten-fold reduction (for the disulfide bicycle) in the IC_50_ [29].

KD2 and other cyclic peptides that specifically bind to the GTP-bound state of oncogenic G12D-RAS-p21 have thus far not been found to inhibit cancer cell growth; this is most likely due to the fact that they do not readily penetrate the cell membranes of these cells [29]. At present, efforts to modify these cyclic peptides with agents that can induce cell membrane penetration are being conducted [29]. It should be noted that cell-penetrating forms of KD2 and related cyclic peptides would be expected to be effective in treating cancer cells that contain only the G12D mutant form of RAS-p21 and may not be effective against RAS-p21 forms that contain other amino acid substitutions at position 12 or at other positions such as at Gly 13 and Gln 61.

### 3.2. Small Molecules Freeze Oncogenic RAS-p21 in an Inactive GDP Bound State 

Another X-ray structure of RAS-p21 with G12C (CysH replaces Gly 12) bound to GDP has also been found to have a “druggable” pocket or groove in a similar domain to the one described above in Section 3.2 immediately above. This pocket seems to further involve the α-helix 3 (residues 89–103) [31,32]. Similar though not identical observations were made for the G12D-K-RAS-p21 mutant [33]. An important feature of this groove is the change in orientation of His 95 in the α-3 helix, enabling compounds with aromatic rings to enter it. Taking the opposite approach to the one used for cyclic peptides in Section 3.1 above, the strategy in this case was to discover small polycyclic molecules that would bind in this pocket in such a way as to lock G12C RAS-p21 in its GDP-binding state, thereby preventing it from exchanging GDP for GTP and disenabling it to activate downstream targets such as RAF. Using compounds from small molecule libraries, several compounds were found that have this effect, as shown in Figure 6.

Of the compounds synthesized and tested, the one with highest affinity, in the picomolar range, for G12C-RAS-p21-GDP was produced at Amgen called AMG510 [34,35,36], now called Sotorasib. This compound is a milestone in targeting RAS after being the first anti-RAS therapy approved by the FDA [35]. Its structure is shown in Figure 6A. In the upper left of this figure, it can be seen that an acryl moiety is present that was designed to react covalently with the -SH group of Cys12 in a Michael addition. This compound has been found not only to bind selectively and covalently to G12C-RAS-p21-GDP, but has been found to block the phosphorylation of ERK in a dose-dependent manner, presumably due to its inability to bind to RAF. AMG-510 has been found to enter cells and it induces impaired cell viability in a number of different cancer cell lines including NCI-H358 non-small cell lung carcinoma and MiaPaCa-2 pancreatic cancer cell lines (IC_50_ = 6 and 9 nM, respectively). It blocks the viability of most cancer cell lines with the G12C mutation except for a lung alveolar cell carcinoma (SW1573) line. It has also been found to cause a reduction in tumor burden in mice with syngeneic G12C-RAS-p21-induced tumors and has been found to eradicate these tumors in a dose-dependent manner. The structure of AMG510 bound to G12C-RAS-p21-GDP is shown in Figure 7. In this figure, the unusual groove involving the switch 2 domain including its α2 helix is shown, and unique to this groove, the α3 helix.

This drug appears to synergize with other anti-cancer agents such as carboplatin, MAPK inhibitors and with immunotherapeutic agents, particularly anti-PD-1 [36]. In this latter case, combination therapy with AMG-510 and anti-PD1 caused dramatic survival rates in CT-26 mice with G12C-RAS-p21 over treatment with either agent alone. In addition, tumors in mice treated with AMG-510 showed a large increase in T cell infiltration, primarily CD8 positive T cells, and seem to have created an inflammatory anti-tumor environment [34,36]. The importance to the overall anti-tumor effect of AMG-510 of a positive anti-tumor immune environment has been shown in recent studies showing that AMG-510 induces only transient tumor regression in immune-deficient nude mice [36].

Of prime importance, as noted above, AMG-510 has received FDA approval for treating G12C-RAS-p21-induced human tumors. At present, the treatment of four patients with NSCCL resulted in 34 and 67% remission in two of the patients and stable disease in the other two patients. A follow-up scan of the second responder patient after 18 weeks of treatment revealed a complete resolution of target remissions. These findings suggest that AMG-510 is a major advance in treating ras-induced human tumors.

However, further studies indicate that human colorectal carcinomas (CRC) with G12C-RAS-p21 mutations do not respond as favorably [36]. Furthermore, a number of mechanisms that result in anti-tumor drug resistance have been identified including the possibility of double mutations in the G12C-RAS-p21 protein that appear to close off the switch 2 pocket, disenabling drug binding. These include mutations at Tyr 96 (Y96D) in the α-3 helix and R68S, H95D or H95Q or H95R (each presumably disenabling the histidine 95 “swing” opening up the switch 2 pocket in G12C-RAS-p21-GDP) [36]. Another drug, RM-018, appears to be able to block this drug resistance [36]. Furthermore, the selective inhibition of the GDP/GTP exchange of G12C-RAS-p21-GDP has been postulated to result in the increased binding of GTP to wild-type RAS-p21, compensating for the inhibition of the mutant protein [33]. In some tumors, the blockade of G12C-RAS-p21-GDP appears to cause an increase in activation of the EGF receptor (EGFR), resulting in the compensatory activation of ras signal transduction pathways [36]. Each of these phenomena are currently being addressed so that the efficacy of AMG-510 and other comparable drugs can be enhanced. As with the KD2 cyclic peptide described above, that may be effective uniquely to G12D-RAS-p21GTP-induced tumors, AMG-510 is expected to be effective only in patients with G12C-RAS-p21-induced tumors.

Other compounds that act in a similar manner to AMG510 have been developed by Mirati Therapeutics including MRTX849, now called Adagrasib, shown in Figure 6B, that has just very recently also been approved by the FDA for treatment of G12C-K-RAS-p21 cancers [33,36]. As can be seen in Figure 6B, MRTX849, like AMG510, is seen to contain a reactive (fluoro) acrylate group attached to a piperazine ring that is attached to a fluorinated polycyclic aromatic compound. A variant of MRTX849, MRTX1133, shown in Figure 6C, has been recently synthesized in which a bicyclo piperazine ring that is positively charged replaced the fluoroacryl group, allowing for this moiety to make favorable electrostatic contact with the Asp 12 residue of G12D-K-RAS-p21-GDP with a binding constant in the low nanomolar range [33,36], making this drug an excellent candidate for the treatment of G12D-K-RAS-p21-induced cancers.

### 3.3. Peptides and Small Molecules from Libraries That Bind to Multiple Forms of RAS-p21-Gpp-NHp

An informatic approach to the design of peptides and small molecules that block RAS-p21-induced cell transformation has been carried out using peptide and small molecule libraries [37]. The X-ray crystal structures of wild-type RAS-p21 and five mutant forms, G12V, G12D, G12C, G13D and Q61H were analyzed for overall similarities in structure and properties. For example, all six forms were computed to have an overall negative surface potential [37]. Then, known anti-cancer peptides from the Cancer PPD database were screened for properties such as high surface positive charge, known strong activity against cancer cell lines and other criteria not listed. A total of 19 peptides were selected for their abilities to dock to specific regions, mainly the P1 loop and the switch 1 and switch 2 domains of the six different RAS-p21 proteins, using commercial servers and programs such as the Haddock server using the PRODIGY program, SwarmDock and FlexPepDock [37].

In the docking procedures for each of the 19 peptides, an “affinity maturation” procedure was further performed in which the docking procedure was re-performed on each peptide in which each residue of the peptide was replaced with all other naturally occurring amino acids. Binding energies for final peptide structures bound to wild-type and mutant forms of RAS-p21 were computed with a formula that uses the number of charge–charge, charge–apolar, polar–polar, polar–apolar interactions between peptide and RAS-p21 and (presumably isolated) apolar residues and charged residues on the peptide and RAS-p21 [37]. The interactions between peptide residues and RAS-p21 residue were computed if the distance between any of the heavy atoms of the two residues were within 5.5 Å. Since the X-ray structure of a RAS-p21 inhibitory peptide bound to this protein was determined [37], the docking procedure was tested on this peptide to determine if its structure bound to the protein could be computed. The resulting computed structure for this peptide was noticeably different from that of the X-ray structure, although it seems to have been docked near the position of the peptide in the X-ray structure.

Concurrently, the ZINC database of small molecules was screened for compounds that had similar structures to GppNHp. This search was further screened by the FAF-Drugs4 server that utilizes a number of pharmacological criteria such as absorption, distribution, metabolites, half-lives, toxicities and other attributes and other “rule of thumb” criteria. Of 7305 structures screened, 474 were used for docking studies against the RAS-p21 protein structures [37].

Of the 19 final peptides, two, called LfcinB (FK**C**RRWQWR**M**KK) and Retro (LG**G**IVSAVKKIV**D**FLG), were computed to have the lowest binding energies and lowest dissociation constants (Kds) to G12D- and G13D-K-RAS-p21 (in the nM range) both at the P1 loop (mutation site) and the switch 1 and 2 domains, the RAF binding sites. Other peptides were computed to have similar but slightly high binding energies and Kds. The lowest energy docked GTP analogues structure from the ZINC database (Zinc 12502230) was computed to have the highest affinity for both structures [37].

Since the both LfcinB and Retro peptides were computed to bind in different regions of the two mutant RAS-p21 proteins, a new peptide containing both peptides was constructed in which the two peptides were connected with a presumably flexible GGGGS connecting or linker peptide. This constructed peptide, Retro-linker-LfcinB, was further modified to contain more positively charged arginine residues, i.e., FK**R**RRWQWRRKK for LfcinB and LG**R**IVSAVKKIV**R**FLG for Retro (substitutions shown in red), to allow for peptide transport across cell membranes [37]. This peptide and the GTP analogue, Zinc 12502230, were concurrently tested on two pancreatic cancer cell lines, MiaPaCa-2 (homozygous for G12C-RAS-p21) and AsPC-1 (homozygous for G12D-RAS-p21) [37].

These two agents were found to reduce the cell viability of both cell lines by about 60 percent at high micromolar concentrations despite their computed dissociation constants being in the nanomolar range [37]. In view of this fact, it would seem important to determine the binding constants of these peptides to the two RAS-p21 forms experimentally. Additionally, at 200 uM peptide concentration, the cell viability of AsPC-1 cells increased from a little less than 40 percent at 100 uM peptide to over 50 percent at 200 uM peptide. The effect of 100 and 200 uM peptide on the MiaPaCa cells was the same, i.e., the inhibitory effect was the same at about 40 percent cell viability [37]. These results suggest that the agents would leave significant residual tumor in patients treated with them. Surprisingly, these agents were not tested on normal cells to determine if they also inhibited wild-type RAS-p21, especially since the structure of this protein was used in the database peptide screening procedure.

As a test for the specificity of the linked peptides and the GTP analogue for inhibiting oncogenic RAS-p21 in cells, their effects on the expression of two downstream targets of RAS-p21, cyclin D and catenin beta-1,encoded by the CCND1 and CTNB1 genes, respectively, were determined [37]. CCND1 expression was reduced in both cell lines, although to a significantly greater extent in MiaPaCa-2 cells. Peculiarly, while these agents almost completely blocked CTNB1 expression in MiaPaCa-2 cells, they increased its expression in the AsPC-1 cells, despite exerting their inhibition of cell viability, suggesting that CTNB1 may not be a reliable marker for ras-21 inhibition [37]. As was performed in the work described above for the KD2 peptide and AMG510, it is desirable to determine if the proposed agents in this study inhibit the binding of oncogenic RAS-p21 to the RBD of RAF oncogenic RAS-p21 and, if so, whether they would also inhibit the binding of wild-type RAS-p21 to the RAF RBD.

## 4. Use of Cancer Cell-Penetrating Peptides Linked to an Anti-Ras Antibody to Inactivate Oncogenic RAS-p21

This rather unique approach [38] is based on the discovery that specific cell-penetrating peptides (CPP) appear to be able to enter cancer cells selectively. These peptides have the basic sequence RAGLQFPVGRLLR, called BR1. Two alternate forms contain repeats of the RLLR sequence: BR2 contains one repeat (RAGLQFPVGRLLRRLLR) while BR3 contains two repeats. These three peptides and a Tat CPP control peptide (RKKRRQRRR) were each labeled with a FITC fluorophore and were incubated with three cancer cell lines (HeLa, human cervical cancer; HCT116, human colon cancer; and B16/F10 mouse melanoma) and three normal cell lines (HaCat, human keratinocytes; BJ, foreskin fibroblasts; and NIH 3T3, mouse fibroblasts) [38].

Significantly, the BR2 peptide was found to enter all three cancer cells but did not appear to enter any of the three normal control cell lines using immunofluorescence, although flow cytometry indicated about 25–30 percent penetrance (as opposed to 100 percent penetrance for the cancer cells) [38]. The control peptide Tat entered all six-cell lines more or less equally. BR3 showed higher levels of entry for tumor cells but did enter normal cells to a lesser extent while BR1 was found not to enter any of the six cell lines. These results suggested that BR2 enters cancer cells with high selectivity. Further investigation of the specificity of BR2 for cancer cells revealed that the ganglioside synthesis inhibitor, PMPP (1-phenyl-2-palmitoyl-3-morpholino-1-propanol), blocks the entry of BR2 into cancer cells but does not block the entry of the tat peptide into these cells [38]. However, the level of ganglioside expression in the membrane fractions of cancer cells versus that of untransformed cells was not determined.

Using the plasmid expression of BR2 linked to the single chain variable fragment or scFv of the anti-RAS-p21 Y13–259 antibody, a fusion polypeptide or BR2-scFv Y13 259 was synthesized as BR2-V_H_-(GGGGS)_3_-V_L_-H_6_ and tested for its ability to kill RAS-p21-transformed tumor cells [38]. The specificity of this fusion polypeptide for cancer cells is provided by the BR2 sequence. Once the polypeptide enters cancer cells, the anti-RAS-p21 mini antibody blocks ras activation.

This construct was tested on HCT 116 (human colon cancer) cells which contain oncogenic substitutions for Gly 13 [38]. The results showed that 67 percent of the cells treated with BR2-scFv underwent apoptosis while about 48 percent of the cells underwent apoptosis when treated with Tat-scFv, and 68 percent of the cells underwent apoptosis when treated with the apoptosis-inducing agent, staurosporine. On the other hand, only 25 percent of the cells treated with scFv alone with no CPP were found to undergo apoptosis [38]. In concurrent experiments, RAS-p21 activity was assayed by measuring its binding to the RBD of RAF. Over a two hour period, 2uM tat-scFv and BR2-scFv caused a 50 and 60 percent drop in RAS-p21 activity. BR2-scFv induced about a 60 percent reduction in cell viability using the MTT assay at 2uM concentration while scFv itself was found to have only a minimal effect [38]. Further evaluation awaits testing of other cancer cells and normal cells since BR2 may enter into normal cells despite this being significantly less than into cancer cells.

## 5. Assessment of Peptide Therapy for Ras-Induced Tumors

Table 1 summarizes each of the peptides and small molecules discussed above that have either been proved to be clinically effective or to have significant potential for treating RAS-p21-induced cancers. Included in this table are the effects of each agent on cancer cells and, importantly, on normal or untransformed cells.

Considering that RAS-p21 has been considered to be “undruggable,” there have been noticeable advances in therapeutic modalities for ras-induced tumors. Careful studies of the X-ray crystal structures of different oncogenic forms of RAS-p21 bound to GDP or GTP have revealed grooves that allow for the insertion of specific (cyclic) peptides or polyaromatic small molecules such as KD2, AMG510 and MRTX 849 (and 1139) that selectively block signal transduction by specific mutations in RAS-p21. AMG510 (Sotorasib) and MRTX 849 (Anagrasib) have already been FDA-approved for the treatment of ras-induced tumors, provided that they have the specific G12C mutation and promise to be effective agents in oncological treatment. If KD2 and similar cyclic peptides can be modified to cross cell membranes, these may prove likewise to be effective agents against G12D-RAS-p21-induced cancers. In this regard, MRTX1139 is another promising drug that blocks G12D-K-RAS-p21 by freezing it in the inactive GDP state. Thus, both KD2 and MRTX1133 target G12D mutant RAS-p21, the former in an inactive GTP-bound state, the latter in the inactive GDP-bound state, suggesting that these agents can act synergistically.

It should be noted that the two FDA-approved drugs, AMG510 and MRTX849, target exclusively G12C-K-RAS-p21. This oncogenic form occurs most frequently in adenocarcinomas of the lung. About 40 percent of ras-induced tumors of this type harbor this mutation [39]. Its occurrence is much less common in all other solid tissue tumors [39]. On the other hand, as a percentage of all ras-induced tumors, G12D-K-RAS-p21 occurs in about 25 percent of lung adenocarcinomas, 37 percent in colon cancers and 50 percent in pancreatic cancers [39]. Thus, targeting this mutant RAS-p21 form is of major concern. In this regard, a major problem in cancer treatment is pancreatic cancer for which the five-year survival remains low (about 5 percent). As it happens, 90 percent of pancreatic cancers contain mutated ras-gene-encoded p21 proteins with substitutions for Gly 12, of which half harbor G12D-KRAS-p21 [39]. Thus, there is a major need for agents like KD2 (provided it can be made to enter cells) and MRTX1133.

On the other hand, a significant number of solid tissue tumors that harbor ras mutations express G12V-K-RAS-p21. As a percentage of all ras-induced tumors, G12V-K-RAS-p21 occurs in about 22 percent of lung adenocarcinomas, 30 percent of colon cancers and 40 percent of pancreatic cancers [39]. Thus, there is a clear need to block this mutant form of RAS-p21, which cannot be treated with agents targeting G12C- and G12D-K-RAS-p21.

As discussed above, other peptide agents and approaches to design additional peptide agents, although less fully developed, show promise in being able to block ras-induced tumor growth irrespective of the mutations at position 12 and other positions in the RAS-p21 sequence. Among these peptides, PNC-1, 2 and 7 peptides linked to the cell-membrane-penetrating leader sequence block oncogenic RAS-p21-induced tumors selectively by interfering with signal transduction pathways that are unique to mutant RAS-p21. It is vital to investigate these pathways further as there appears to be “escape” pathways that allow for the wild-type RAS-p21 protein to function normally in normal cells. These agents induced either the phenotypic reversion of cancer cells heterozygous for mutant RAS-p21 or tumor cell necrosis for cancer cells that are homozygous for mutant RAS-p21. None of these peptides have a demonstrable effect on the viability or growth of untransformed cells. These peptides remain to be tested in vivo.

The immunological approach involving the attachment of the anti-ras Y13 259 scFv heavy and light chain segments to the tumor cell membrane recognition peptide, BR2, seems to be another productive approach in the design of specific anti-RAS-p21 agents. Using this agent, specificity for cancer cells is provided by the BR2 peptide that allows for the anti-RAS-p21 antibody to enter ras-transformed cancer cells. Again, in vivo studies with this construct remain to be performed.

Finally, much of the successful development of anti-ras peptides and small molecules has resulted from library searches for these molecules. These were involved in the development of AMG510 and MRX849 and KD2, the latter resulting from a combination of library searches, use of expression vectors to produce peptides and use of cyclization methods to produce the cyclic peptides. The screening procedures for all of these peptides and small molecules are based upon the ability of these agents to bind to specific forms of oncogenic RAS-p21. In view of the need for more general anti-ras agents, the use of informatic approaches such as the ones described in Section 3.3 above will provide an essential computer-based analysis of peptide constructs, leading to therapeutic advances in targeting ras-induced tumors.

## Figures and Tables

**Figure 1 biomedicines-11-00471-f001:**
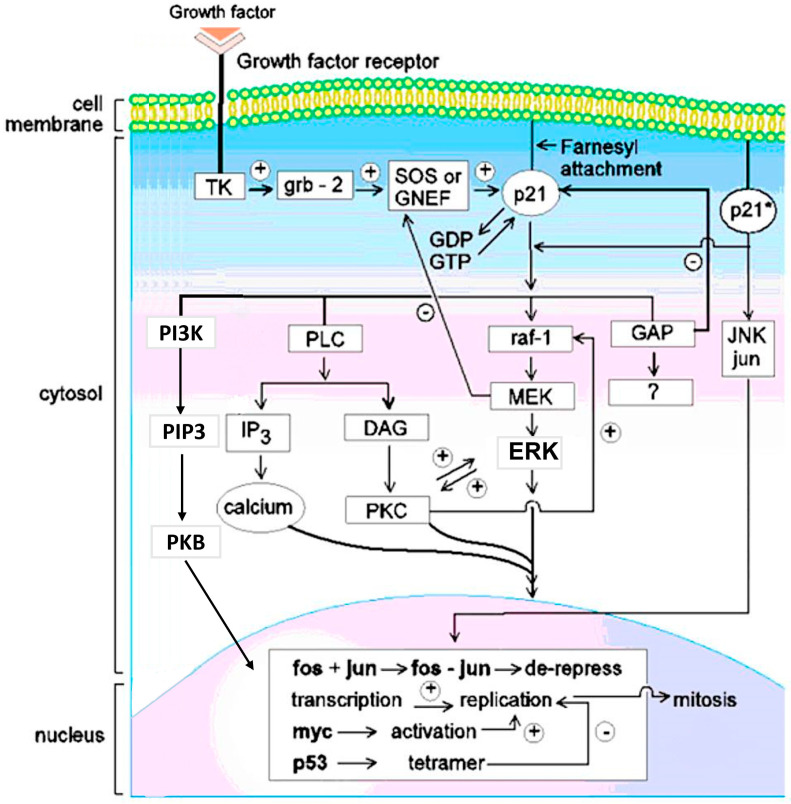
**Pincus et al. Wild-type (RAS-p21) and Oncogenic (RAS-p21*) Signal Transduction Pathways.** A growth factor such as EGF or insulin binds to its receptor (upper left)—for example, EGFR—extracellularly. The receptor contains three domains: the extracellular growth factor-binding domain, the trans-membrane domain and the intracytosolic domain. The latter contains a tyrosine kinase (TK in the figure) which becomes activated when the receptors dimerize upon binding to the growth factor. This event enables the intracytosolic domain to bind to the adapter grb-2 protein that concurrently binds to and activates a guanine nucleotide exchange factor (GNEF), SOS, a protein that in turn binds to RAS-p21, causing it to exchange GDP for GTP. In the GTP-bound state, RAS-p21 becomes activated. In its activated state, it induces mitogenic signal transduction which occurs only if RAS-p21 is bound to the inner surface of the cell membrane through a covalently attached farnesyl moiety in thioether linkage to Cys 186 catalyzed by the enzyme, farnesyl transferase. If RAS-p21 is not membrane-bound, it acts as a competitive inhibitor of membrane-bound ras. An important target of activated membrane-bound RAS-p21 in mitogenic signaling is the serine-theonine kinase, raf, a 74 kDa Ser/Thr kinase protein that, in turn, directly binds to and activates another serine-threonine kinase, MEK in a so-called kinase cascade. The latter protein activates ERK. This all-important protein shuttles between the cytosol, where it is involved in cytoskeletal rearrangements (the name MAPK stands for mitogen activated or microtubule-activating protein kinase) and the nucleus in which it activates the critically important nuclear transcription factor, *fos,* that itself forms a heterodimeric complex, called AP1, with the critically important nuclear transcription factor, *jun,* that is activated by another kinase, jun-N-terminal kinase (JNK), that is normally activated on a separate pathway called the stress-activated protein (SAP) pathway. The AP1 complex induces the transcription of many pro-mitogenic proteins including cyclins and nuclear skeletal proteins called nuclear matrix proteins ( or NMP’s); other nuclear proteins, such as *myc*, also transcriptionally active, are also often activated in this process. Anti-oncogene proteins, such as p53, also become activated. This protein blocks the transcription of pro-mitotic proteins (minus sign on p53 arrow) and induces apoptosis in transformed cells. On the right of the figure, oncogenic RAS-p21 (RAS-p21*) is shown interacting with and activating JNK, which seems to be a unique aspect to oncogenic RAS-p21. Work in oocytes (see text) suggests that wild-type RAS-p21 may also activate the dual specificity kinase TOPK (not shown in the figure), which is known to bind to RAF and is in the MEK superfamily. Apparently, oncogenic RAS-p21 does not utilize this pathway. The activated state of RAS-p21 is regulated by the protein, GAP (GTPase-Activating Protein), that enhances the native GTPase activity of RAS-p21. GAP induces the hydrolysis of GTP to GDP, resulting in the inactivation of RAS-p21. However, GAP itself may be involved in mitogenic signal transduction, hence, the question mark in the arrow below the GAP box in the figure. As shown on the left side of the figure, RAS-p21 also interacts directly with phosphoinositol-3-hydroxy kinase (PI3K), which catalyzes the synthesis of phosphatidylinositol 3-hydroxyphosphate or triphosphate (PIP3) that is involved in the activation of protein kinase B (PKB), also known as AkT, of which among many functions, induces cell proliferation. Activated RAS-p21 further induces the activation of phospholipase C (PLC) that catalyzes the formation of diacylglycerol (DAG), an activator of protein kinase C (PKC) that is especially critical to the oncogenic *RAS-p21* pathway and the synthesis of inositol triphosphate (IP3) of which mobilizes the release of calcium.

**Figure 2 biomedicines-11-00471-f002:**
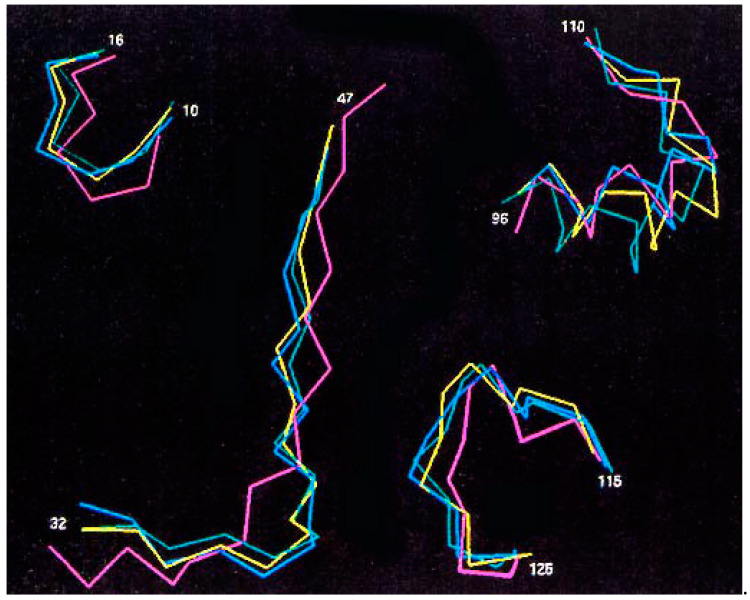
**C^α^ traces for effector domains of molecular dynamics average structures of different forms of RAS-p21.** Color scheme is: pink, wild-type RAS-p21-GDP; yellow, RAS-p21-GTP; blue, G12V-RAS-p21-GTP; green, Q61L-RAS-p21-GTP. The residue numbers for each domain are shown in the figure [3].

**Figure 3 biomedicines-11-00471-f003:**
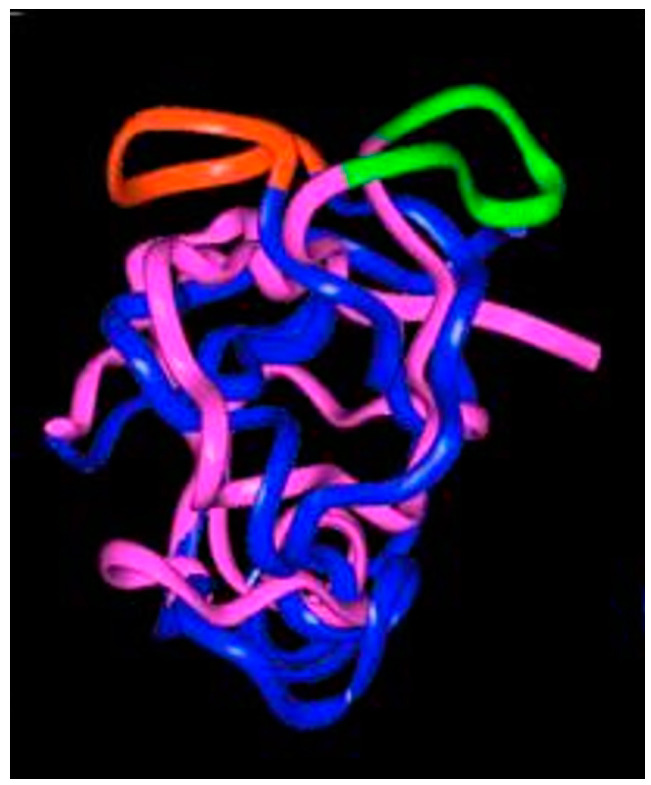
Ribbon representation of the superimposed average molecular dynamics RAF RBD structures bound to oncogenic (Val 12-) RAS-p21 (blue and red structure) and to wild-type RAS-p21 (pink and green). The exposed loop on the top surface of the figure of the 97–110 segment is colored green in the wild-type complex and red in the oncogenic complex (adapted from [15]).

**Figure 4 biomedicines-11-00471-f004:**
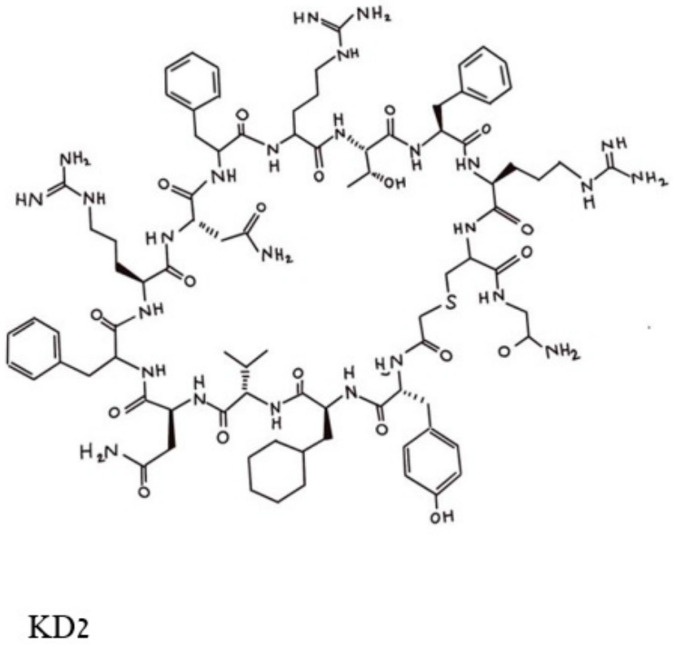
Pincus et al. Structure of the cyclic peptide, KD2, that blocks the activation of the G12D-RAS-p21-GTP complex. The Thr 10 residue whose side chain -OH group makes contact with the Asp 12 residue is in the top middle of the figure and can be identified by the -OH in this area.

**Figure 5 biomedicines-11-00471-f005:**
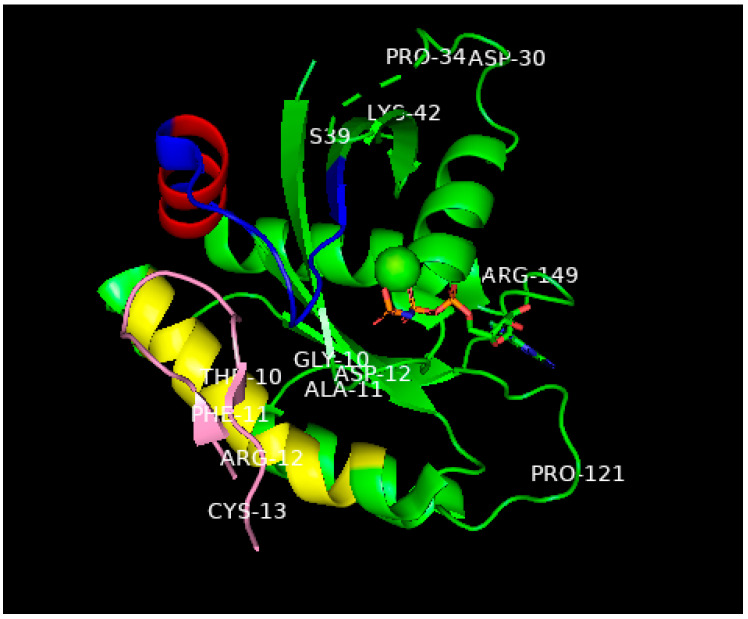
Ribbon representation of the X-ray structure (ref. [29], PDB entry 6WGN) of KD2 bound to G12D-RAS-p21-GppNHp rotated to show the location of KD2 (pink) in the groove in the switch 2 domain (residues 55–76) which is colored blue except for the α2 helix involving residues 67–73, colored red. This helix is rotated by 90° relative to its position in wild-type RAS-p21. The yellow-colored segment is the α3 helix involving residues 89–103. A significant segment of the pink KD2 peptide is seen to bind in a deep groove between the α2 helix and a significant part of the switch 2 domain shown as the blue strand to the right and above the pink segment. GppNHp is seen in the middle, right of the figure with the guanine ring on the far right; the β- and ϒ-phosphates (the phosphates are colored orange) are seen to bind to the magnesium ion shown as a green sphere in the center of the figure. The oncogenic Asp 12 residue is shown in the middle of the P loop domain, some of whose other residues (Gly 10 and Ala 11) are also shown. Additionally, to be noted: The all-important switch 1 domain containing residues 32–47 that contacts the RBD of RAF is shown in the upper right of the figure. Some of the residues in this domain such as Pro 34, Ser 39 and Lys 42 are shown to help identify this domain. Between Pro 34 and Ser 39, there is a series of green dashes denoting the absence of location of the atoms in the intervening segment, this is due to the disordered switch 1 domain in state 1 (inactive) in GTP-bound RAS-p21. See text for explanation.

**Figure 6 biomedicines-11-00471-f006:**
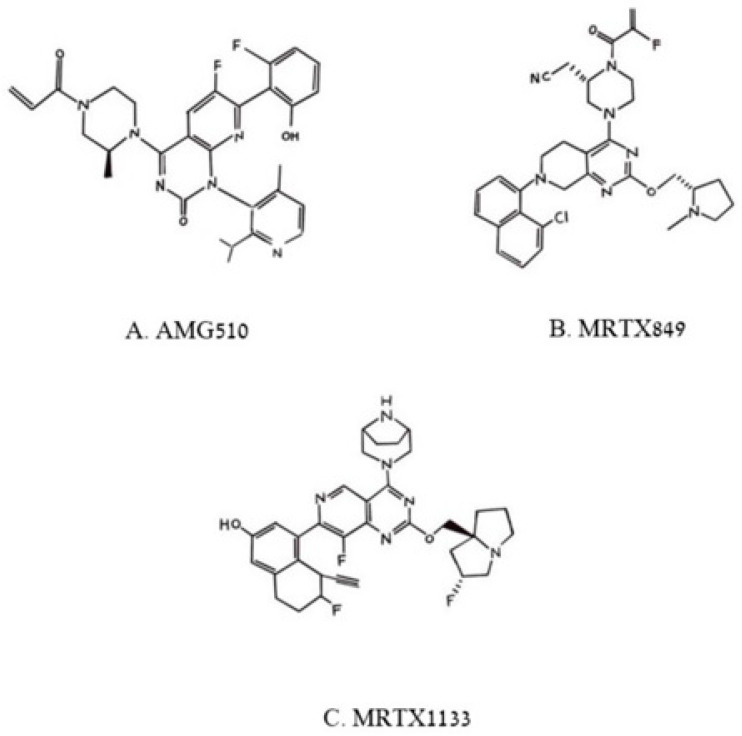
Pincus et al. Chemical structures of three poly aromatic ring molecules, AMG500 (**A**), MRTX849 (**B**) and MRTX 1139 (**C**) that insert into a groove formed by the switch 2 domain, its α2 helix and its α3 helix. The first two compounds, (**A**,**B**), bind uniquely to G12C-K-RAS-p21-GDP, the third, (**C**), binds to G12D-K-RAS-p21-GDP.

**Figure 7 biomedicines-11-00471-f007:**
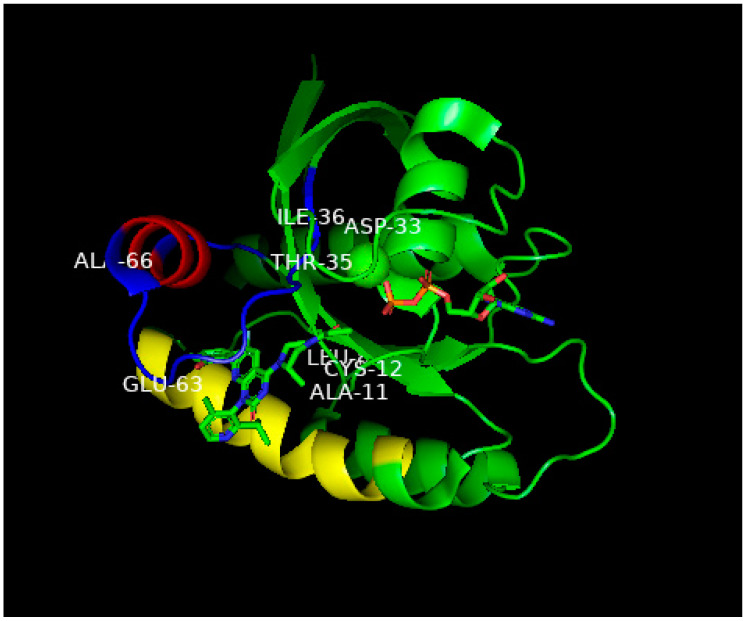
Ribbon representation of the X-ray structure of AMG510 bound to G12C-RAS-p21-GDP from Ref. [36]. Coordinates were obtained from the Protein Data Base (PDB) identity number 7RPZ and rotated to give a clear view of the binding pocket occupied by AMG510. The color scheme for critical domains is the same as for Figure 5. G12C-RAS-p21 uniquely has a groove formed by the switch 2 domain, consisting of residues 55–76 (colored blue except for its α2 helix that is colored red) and its α3 helix (residues 89–103, colored yellow). The orientation of this complex is similar to that of the KD2-G12D-RAS-p21-GppNHp complex shown in Figure 5. As in Figure 5, the nucleotide (here, GDP) is oriented horizontally such that the guanine ring is on the far right in the middle of the figure, and a β-phosphate oxygen binds to the magnesium ion (green sphere in the middle of the figure). The AMG510 molecule can be seen in the lower left of the protein inserted into a groove between the yellow α3 helix and the blue switch 2 loop region above it in the figure. Just above Leu 80 (“Leu” in the middle of the figure), there is a red projection that is the carbonyl oxygen of the acryl moiety of the drug that is positioned to react with the-SH group of Cys 12 labeled in the figure. In contrast to Figure 5, the switch 1 domain is seen to be well-defined and can be identified by the labeled residue positions Asp 33, Thr 35 and Leu 36. Unlike the positioning of switch 1 residues on the surface of the protein in Figure 5, the switch 1 residues of G12C-RAS-p21-GDP are not as exposed.

**Table 1 biomedicines-11-00471-t001:** Peptides and Small Molecules that Block Oncogenic RAS-P21.

Source of Peptide or Small Molecule	Amino Acid Sequence or Small Molecule	Site of Action	Effect on Cancer Cells	Effect on Normal Cells
RAS-p21,35–47 (PNC-7) ^1^	TIEDSYRKQVVID and TIEDSYRKQVVID-Leader ^2^	G12V-H-RAS-p21 binding to RAF	Leader form induces phenotypic reversion of rat TUC-3 pancreatic cancer cells and HT1080 human fibrosarcoma cells; causes 100% cell death of MIA-PaCa-2 human pancreatic cancer cells.	Leader form has no effect on rat BMRPA1 pancreatic acinar cells and human keritinocytes.
RAS-p21,96–110 (PNC-2) ^1^	YREQIKRVKDSDDVP and YREQIKRVKDSDDVP-Leader ^2^	G12V-H-RAS-p21 binding to JNK, SOS	Leader form induces phenotypic reversion of rat TUC-3 pancreatic cancer cells and HT1080 human fibrosarcoma cells; causes 100% cell death of MIA-PaCa-2 human pancreatic cancer cells.	Leader form has no effect on rat BMRPA1 pancreatic acinar cells and human keritinocytes.
RAS-p21,115–126 (PNC-1) ^1^	GNKCDLAARTVE and GNKCDLAARTVE-Leader ^2^	G12V-H-RAS-p21 binding to JNK, SOS	Leader form induces phenotypic reversion of rat TUC-3 pancreatic cancer cells and HT1080 human fibrosarcoma cells; causes 100% cell death of MIA-PaCa-2 human pancreatic cancer cells.	Leader form has no effect on rat BMRPA1 pancreatic acinar cells and human keritinocytes.
RBD of RAF 97–110 ^1^	AVFRLLHEHKGKKA	G12V-H-RAS-p21 binding to RAF and MEK	NT ^3^	NT
SOS 994–1004 ^1^	LNPMGNSMEKE	G12V-H-RAS-p21 binding to SOS	NT	NT
GST-π 34–50 ^1^	TIDTWMQGLLKPT CLYG	G12V-H-RAS-p21 binding to JNK/jun	NT	NT
GST-π 169–182 ^1^	CLDNFPLLSAYVAR	G12V-H-RAS-p21 binding to JNK/jun	NT	NT
CPP-RAS-p21 169–188 (Mut3DPTSh1)	*VKK-KIKAEIKI*-**KMSKDGKKKKKKSRTRCTVM** CPP (Cell-Penetrating Peptide) is the italicized sequence	K-RAS-p21 binding to RAF	Apoptosis in MDA-MB-231 and HBCx17 human breast cancer, SW626 human ovarian cancer and SW480 human colon cancer. Lower viability of H1650, HBEC human NSCCL. Minimal effect on HCT-116 and HT-29 human colon cancer cell	NT
High diversity cDNA Library	KD2 Cyclic Peptide	Binds to crevice in G12D-K-RAS-p21 bound to GppNHp blocking its activation	Peptide blocks binding specifically of G12D-K-RAS-p21 to RAF in vitro, but peptide does not enter cells.	Peptide blocks binding specifically of G12D-K-RAS-p21 to RAF in vitro, but peptide does not enter cells.
Polyaromatic molecule library	AMG510	Binds to crevice in G12C-K-RAS-p21-GDP maintaining it in the inactive state	Induces cell death in CT26, MiaPaCa-2 human pancreatic cancer cells, NCI-H358 human NSCCL, CT26 murine colorectal ca in culture and in mice. FDA approved for treatment of human cancers with G12C-K-RAS-p21-induced tumors. Treatment of 4 patients with NSCCL treatment of four patients with NSCCL resulted in 34 and 67% remission in two of the patients and stable disease in the other two patients.	
Polyaromatic molecule library	MRTX849	Binds to crevice in G12D-K-RAS-p21-GDP maintaining it in the inactive state	MRTX 849 Cell Lines: cytotoxic to human NSCCL, H2030, H358, 1573, 1792 cells. In vivo: Eradicates Colorectal-CR6243 Lung-Calu-1 (KRAS-G12C) Pancreas-MIA PaCa-2 (KRASG12C Lung-LU65 Lung-H1373. FDA approved for human cancers with G12C-K-RAS-p21-induced tumors. FDA approved for treatment of human cancers with G12C-K-RAS-p21-induced tumors.	
Cancer PPD Data Base, docking programs including PRODIGY, SwarmDock and FlexPepDock. 1st black sequence is Retro peptide, blue sequence is linker, and last black sequence is LfcinB peptide. Red R (Arg) residues were substituted to facilitate cell penetration.	LGRIVSAVKKIVRFLGGGGGS-FKRRRWQWRRKK	Docks to known K-RAS-p21 site exposed in wt-, G12V-,G12D-, G12C-,G13D-Q61H-K-RAS-p21.	Reduces viability (by about 6-%) of MiaPaCa-2 and AsPC-2 human pancreatic cancer cells.	NT
Construct peptide using BR2 as cancer cell CPP linked to the variable heavy chain domain of the anti-ras MAb, Y13 259. This is linked in turn to a linker sequence that is linked to the variable light chain that is attached to the hexaHis sequence.	BR2-V_H_-(GGGGS)_3_-V_L_-H_6_ BR2 = RAGLQFPVGRLLRRLLR.	Enters cancer cells selectively due to BR2; the scFv is directed against ras-21 and blocks its mitogenic signaling.	Induced 60 percent apoptosis of HCT-116 human colon cancer cells.	NT

^1^ These peptides were found to inhibit oncogenic G12V-H-RAS-p21-induced meiotic division in Xenopus oocytes but not insulin-activated wt-H-RAS-p21-induced meiotic division. ^2^ Leader sequence = KKWKMRRNQFVKVQRG. ^3^ NT = Not Tested.

## Data Availability

Not applicable.

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
