# Peer review of "Peptides That Block RAS-p21 Protein-Induced Cell Transformation"

_biomedicines, 2023, doi:10.3390/biomedicines11020471_

Round 1
Reviewer 1 Report
The article: “Peptides that Block ras-p21 Protein-Induced Cell Transformation” is a review based on a small pool of references. Many other references can be cited. The references were presented in too many details, but the focus was lost.
In my opinion, after major revisions listed below this paper will be acceptable for publication in “Biomedicines”.
1. The Figures 4. and 6. should be drawn again because they are of very bad quality and the structure of compound KD2 is not correct. In Gly-amide, it should be double bond to oxygen. The important amino acids such as Thr in compound KD2 should be pointed in different color.
2. What are Sections A, B, C (IA, IIA…) that the authors constantly refer to? I did not find these sections.
3. Line 206 Instead “…G12R, G12V, Q61His, Q61L…” it should be …G12R, G12V, Q61H, Q61L… Why is His different from other amino acids?
4. Line 570-571 In the sentence: “…bridge, have been synthesized and found to block the binding of G12D-ras-p21-GppNHp to the raf RBD with tenfold reduction (for the disulfide bicycle) in the IC50 (29).” it is not clear if the disulfide has a tenfold higher effect.
5. uM should be µM
6. wrong sign for the degree was used
7. Table 1 gives a good overview of this paper, but it is necessary to put ref. for every entry.
8. The spaces between words in the text should be checked.
Author Response
Please see the attachement.

Reviewer 2 Report
The review article is quite comprehensive, and the authors do a very thorough job of describing the ras system and current state of the field regarding inhibitors, especially peptide inhibitors. This article, when published, will be a great resource for those in the field.
Several points the authors should consider in revisions:
1. A significant amount of discussion is focused on the CPP-fused molecules. As such, the article would benefit from a more thorough discussion of CPPs on the whole, not just the specific CPP used in the studies review. One paragraph regarding the overview and background of CPPs would be sufficient in my opinion.
2, Several of the figures could use improvement for readability/visibility. Figure 3 and 6 seem low resolution and should be improved. Figure 5 would benfit from changing the color of the Mg ion which blends in with the background (and it's not obvious why S39 is not hyphenated, consistent with the other resiudes).
3. Table 1 has so much valuable information, but it difficult to read considering the size of the columns & cells, spilling over onto multiple pages. I do not have a direct suggestion, but ask that the authors think about alternate ways to present this. If a reasonable alternative is found, i encourage it considering the value of the information.
4. Table one has a number of minor typos (in vivo not italicized, linked incorrectly spelled "ilnked").
Author Response
Please see attachement.

Reviewer 3 Report
Although I find this review interesting and informative about recent research on RAS targeting, there are several important points that should be addressed before publication.
Major points:
1- Section1 (background) is full of errors and should be reviewed and completed. There are mistakes in signaling pathways, as well as misnamed proteins, etc. (see some comments in the attached file). In addition, Figure 1 must be improved. I would suggest the authors to make a completely new figure and avoid using a figure that has mistakes.
2- Many references are missing in this work. This is indicated in the comments on the attached file. In addition, is very non-conventional to add references to the title, and this is observed in several sections. I would suggest to remove all references from the titles and if a section includes work from only one source, use an introductory sentence to establish this and to add the reference.
3- Use a proper nomenclature for genes and proteins. Many non-conventional names are used throughout the manuscript and this should be avoided.
4- The paragraph from 467 to 483 must be eliminated. A review is not an appropriate setting to defend one's work, as mentioned in the attached file there are other options more appropriate for this.
Minor points:
See comments in the attached file. In addition review the manuscript for errors on miss-spelling and spacing.

Author Response
Please see attachement.

Round 2
Reviewer 1 Report
The authors did not answer my questions and suggestions.
Reviewer 3 Report
I have carefully read the answers to my suggestions on the work entitled "Anti-cancer Peptides and Peptide-Like Molecules" as well as the changes to the reviewed manuscript, and find that the new version answers many of the concerns I found in the first version submitted for publication. However, there is one change that I suggested in my first review that in the authors' answers they state that it has been changed, but there are no changes from the original version. This is Figure 1. The authors say in their answer: "This figure has been corrected. please see Figure 1. The legend to this figure has also been changed correspondingly.": The legend has been changed, but the figure is the same as before. Please, check this and change this figure to correct the mistakes that have been solved in the figure legend.
I would ask the authors to avoid sarcasm and just answer or reject the suggestions. As a reviewer I try not only to analyze the eligibility of a manuscript for publication, but also to help the authors to improve the manuscript. Most of my comments are suggestions in that line (minor comments), and those can be accepted or rejected with an explanation (as in some of the suggestions of my original review here). For that reason, words such as "nitpicking" or sarcasm as in "As far as JNK is concerned, we call it-surprise!-JNK." are unnecessary and unpolite, thank you.
Round 3
Reviewer 1 Report
The paper can be published in Biomedicines.
Reviewer 3 Report
This last version of the manuscript is ready for publication.